# Impact of Early Intrapatient Variability of Tacrolimus Concentrations on the Risk of Graft-Versus-Host Disease after Allogeneic Stem Cell Transplantation Using High-Dose Post-Transplant Cyclophosphamide

**DOI:** 10.3390/ph15121529

**Published:** 2022-12-09

**Authors:** Daniel N. Marco, María Queralt Salas, Gonzalo Gutiérrez-García, Inés Monge, Gisela Riu, Esther Carcelero, Joan Ramón Roma, Noemí Llobet, Jordi Arcarons, María Suárez-Lledó, Nuria Martínez, Alexandra Pedraza, Ariadna Domenech, Laura Rosiñol, Francesc Fernández-Avilés, Álvaro Urbano-Ispízua, Montserrat Rovira, Mercè Brunet, Carmen Martínez

**Affiliations:** 1Hematopoietic Stem Cell Transplantation Unit, Hematology Department, Institute of Hematology and Oncology, Hospital Clínic, IDIBAPS, 08036 Barcelona, Spain; 2Department of Pharmacy, Pharmacy Service, Hospital Clínic, IDIBAPS, 08036 Barcelona, Spain; 3Pharmacology and Toxicology Laboratory, Biochemistry and Molecular Genetics Department, Biomedical Diagnostic Center, Hospital Clínic, IDIBAPS, 08036 Barcelona, Spain

**Keywords:** tacrolimus, therapeutic drug monitoring, graft-versus-host disease, tacrolimus intra-patient variability, allogeneic stem cell transplantation, post-transplant cyclophosphamide

## Abstract

Tacrolimus (Tac) is a pivotal immunosuppressant agent used to prevent graft-versus-host disease (GVHD) after allogeneic stem cell transplantation (alloHSCT). Tac is characterized by a narrow therapeutic window and a high inter-patient and intra-patient pharmacokinetic variability (IPV). Although high IPV of Tac concentrations has been associated with adverse post-transplant outcomes following solid organ transplantation, the effects of Tac IPV on alloHSCT recipients have not been determined. Tac IPV was therefore retrospectively evaluated in 128 alloHSCT recipients receiving high-dose post-transplant cyclophosphamide (PTCy) and the effects of Tac IPV on the occurrence of acute GVHD (aGVHD) were analyzed. Tac IPV was calculated from pre-dose concentrations (C_0_) measured during the first month after Tac initiation. The cumulative rates of grades II-IV and grades III-IV aGVHD at day +100 were 22.7% and 7%, respectively. Higher Tac IPV was associated with a greater risk of developing GVHD, with patients having IPV > 50th percentile having significantly higher rates of grades II-IV (34.9% vs. 10.8%; hazard ratio [HR] 3.858, *p* < 0.001) and grades III-IV (12.7% vs. 1.5%; HR 9.69, *p* = 0.033) aGVHD than patients having IPV ≤ 50th percentile. Similarly, patients with IPV > 75th percentile had higher rates of grades II-IV (41.9% vs. 16.5%; HR 3.30, *p* < 0.001) and grades III-IV (16.1% vs. 4.1%; HR 4.99, *p* = 0.012) aGVHD than patients with IPV ≤ 75th percentile. Multivariate analyses showed that high Tac IPV (>50th percentile) was an independent risk factor for grades II-IV (HR 2.99, *p* = 0.018) and grades III-IV (HR 9.12, *p* = 0.047) aGVHD. Determination of Tac IPV soon after alloHSCT could be useful in identifying patients at greater risk of aGVHD.

## 1. Introduction

Allogeneic stem cell transplantation (alloHSCT) is a potentially curative therapy for a variety of malignant and non-malignant hematological disorders. Both acute and chronic graft-versus-host disease (GVHD) remain the major clinical complications of alloHSCT, limiting survival and quality of life; therefore, preventing GVHD is critical for transplant success [1]. The calcineurin inhibitors cyclosporine and tacrolimus (Tac), in combination with methotrexate or mofetil mycophenolate (MMF), have been the cornerstone of immunosuppressive therapy in alloHSCT for decades [2,3]. GVHD rates have been markedly reduced following the introduction of new agents, such as antithymocyte globulin (ATG) and high-dose post-transplant cyclophosphamide (PTCy) [4,5,6,7,8]. Specifically, PTCy plus Tac and MMF is currently a successful and widely utilized combination for haploidentical HSCT (haploHSCT), as well as in patients receiving transplants from HLA-identical sibling (MSD) and matched unrelated donors (MUD).

Tac is characterized by a narrow therapeutic window and high inter- and intra-individual pharmacokinetic variability [9,10]. In current clinical practice, therapeutic drug monitoring (TDM) by measuring Tac trough level (C_0_) is used to maintain concentrations within therapeutic ranges. Although the optimal blood concentration of Tac has not been determined, C_0_s of 10–20 ng/mL [11,12] and 5–15 ng/mL [13,14] have been recommended as therapeutic targets when Tac is combined with methotrexate and sirolimus, respectively. Less is known, however, regarding its optimal C_0_ when Tac is combined with PTCy. Moreover, although higher maintained Tac blood concentrations have been associated with Tac toxicity and transplant-related mortality (TRM), no consensus exists regarding the impact of low Tac concentrations on the incidence of GVHD [11,12,15,16,17].

New TDM strategies are required to assess the risk of GVHD in individuals. Measuring the intrapatient variability (IPV) of Tac concentrations has been reported as useful for optimizing immunosuppressive therapy in solid organ transplant recipients [18,19], as significant variations in Tac C_0_ may lead alternatively to periods of underexposure and overexposure, resulting in higher risks of organ rejection and drug toxicity, respectively. Thus, higher Tac IPV has been associated with poor outcomes, such as acute rejection, graft loss, and mortality. These deleterious effects have been reported in recipients of renal, lung, heart, and liver transplants [20,21,22,23]. By contrast, the effects of Tac IPV on outcomes have not been evaluated in alloHSCT recipients.

The present study evaluated Tac IPV based on C_0_ measurements in the early period after alloHSCT in patients receiving PTCy-based GVHD prophylaxis regimens and analyzed the effects of Tac IPC on the incidence of acute GVHD (aGVHD).

## 2. Results

### 2.1. Baseline Characteristics of the Patients

The characteristics of the 128 alloHSCT recipients and the transplant procedure are summarized in Table 1. The median patient age was 49 years (range, 18–69 years), and 72 (56%) patients were male. The most frequent diagnosis was acute leukemia or myelodysplastic syndrome, observed in 89 (70%) patients. In total, 79 (62%) patients underwent reduced intensity conditioning (RIC) alloHSCT, 82 (64%) received grafts from unrelated donors, and 125 (98%) received peripheral blood as the stem cell source. Further, 94 (73%) patients received PTCy and Tac as GVHD prophylaxis, and 34 (27%) received PTCy, Tac, and MMF. Oral Tac formulations (BID and QD) were used mainly in RIC alloHSCT (65.8%).

### 2.2. Tacrolimus IPV, TISS, and C_0_ Levels

Median Tac IPV was 46% (interquartile range (IQR) 32–56%). Patients with Tac IPV below the 25th percentile (31 patients) were considered to have very low variability, whereas those with IPV above the 75th percentile (31 patients) were considered to have very high variability. Comparisons of the three Tac dose regimens showed that median Tac IPV was significantly lower for intravenous (31%) than for oral Tac, either BID (47.5%, *p* = 0.004) or QD (51.5%, *p* < 0.001), with no difference between the two oral regimens. Analysis of Tac C_0_ according to Tac IPV showed that the lower the IPV, the faster the therapeutic range was reached. Thus, 82% of patients with Tac IPV below the 25th percentile achieved therapeutic range at 48 h, compared with 51% of patients with IPV over the 25th percentile (*p* = 0.001). Conversely, 32% of patients with Tac IPV over the 75th percentile achieved therapeutic levels within the first 48 h compared with 68% of patients with IPV below the 75th percentile (*p* < 0.001).

Although 30% of patients had sub-therapeutic Tac C_0_ (<5 ng/mL) at TISS, only 3%, 6%, 2%, and 3% of patients had sub-therapeutic Tac C_0_ at 7, 14, 21, and 28 days, respectively (Figure 1). Conversely, despite close C_0_ monitoring and dose adjustments, 11% of patients had supra-therapeutic Tac C_0_ (>15 ng/mL) at TISS, whereas 27%, 17%, 20%, and 23% had supra-therapeutic Tac C_0_ at 7, 14, 21, and 28 days, respectively. Evaluation of the three Tac regimens showed that Tac levels were significantly higher at TISS with intravenous than with oral Tac. The proportion having Tac C_0_ < 5 ng/mL was significantly lower in patients receiving intravenous (8.3%) than oral Tac, either BID (31.3%, *p* = 0.028) or QD (41.7%, *p* < 0.001), and the proportion having C_0_ > 15 ng/mL was significantly higher in patients receiving intravenous (28%) than oral Tac, either BID (3%, *p* = 0.007) or QD (5%, *p* < 0.001). No significant differences were observed between the oral BID and QD Tac at any time.

Univariate and multivariate analysis of patient’s and transplant’s baseline characteristics that can influence Tac IPV and TISS are displayed in Appendix A.

### 2.3. Impact of Tacrolimus IPV, TISS, and C_0_ on Acute GVHD

Overall, the estimated cumulative incidences of grades II-IV and grades III-IV aGVHD at day +100 were 22.7% (95% confidence interval (CI) 15.8–30.3%) and 7% (95% CI 3.4–12.3%), respectively. The median time from day 0 to aGVHD onset was 38 days (range 17–89 days).

The risk of developing aGVHD was higher in patients with a higher Tac IPV (Table 2, Figure 2). The incidence of grades II-IV aGVHD was higher in patients with Tac IPV above than below the 25th percentile (26.9% vs. 11.8%, *p* = 0.088), although this difference was not statistically significant (Figure 2A). Patients with Tac IPV above the 50th percentile had significantly higher rates of grades II-IV (34.9% vs. 10.8%, *p* = 0.0018) and grades III-IV (12.7% vs. 1.5%, *p* = 0.033) aGVHD than patients with Tac IPV below the 50th percentile (Figure 2B). Furthermore, patients with very high variability (IPV above the 75th percentile) had significantly higher rates of grades II-IV (41.9% vs. 16.5%, *p* < 0.001) and grades III-IV (16.1% vs. 4.1%, *p* = 0.012) aGVHD than patients with Tac IPV below the 75th percentile (Figure 2C).

The cumulative incidence of grades II-IV aGVHD at 100 days was significantly higher for patients with Tac TISS < 5 ng/mL than for patients with Tac TISS ≥ 5 ng/mL (37.8% vs. 16.7%, *p* < 0.0067) (Table 2, Figure 3). Although the percentage of patients developing grades III-IV aGVHD was also higher in those with TISS < 5 ng/mL than TISS ≥ 5 ng/mL (10.5% vs. 5.6%), the difference was not statistically significant. Tac C_0_ at 7 days <5 ng/mL was also associated with a higher risk of grade II-IV aGVHD (Table 2). Thus, all patients with C_0_ < 5 ng/mL at 7 days developed aGVHD, whereas the cumulative incidence in patients with Tac C_0_ ≥ 5 ng/mL at 7 days was 20.2% (*p* < 0.001). Tac C_0_ at days 14, 21, and 28, however, had no impact on GVHD risk (Table 2).

Other risk factors for aGVHD are summarized in Table 2 (univariate) and Table 3 (multivariate). There were no differences between intravenous and oral (BID and QD) Tac formulations and aGVHD incidence or severity (Table 2). Multivariate analysis showed that high Tac IPV was an independent risk factor for aGVHD (Table 2). Thus, patients with Tac IPV over the 50th percentile had higher probabilities of being diagnosed with grades II-IV (HR 2.99, 95% CI 1.21–7.39, *p* = 0.018) and grades III-IV (HR 9.12, 95% CI 1.03–80.56, *p* = 0.047) aGVHD than patients with Tac IPV below the 50th percentile. Although univariate analyses showed that IPV > 25th percentile and >75th percentile were associated with higher risks of aGVHD, these IPV cut-off values were not included in the multivariate analysis because they were mutually exclusive. Multivariate analysis models for IPV > 25th percentile and >75th percentile are shown as Appendix A. In addition, patient age (HR 1.11, 95% CI 1.05–1.18, *p* < 0.001) and the use of MAC regimens (HR 9.94, 95% CI 2.84–34.77, *p* < 0.001) were associated with significantly higher risks of clinically relevant aGVHD.

### 2.4. Impact of Tacrolimus IPV, TISS, and C_0_ on Toxicity

During the study period, 97 (76%) of the 128 patients experienced at least one episode of acute kidney injury (AKI), with 58 (45%) having moderate or severe AKI (grades 2–3). Maximum creatinine concentrations increased from a median of 0.76 mg/dL (IQR, 0.55–0.97 mg/dL) before alloHSCT to a median of 1.61 mg/dL (IQR 0.90–2.32 mg/dL) during follow-up, a difference corresponding to a relative increase of 2.1 in the maximal/basal serum creatinine (mCr/bCr) ratio. Supratherapeutic levels of Tac C_0_ (>15 ng/mL) at different time points after transplant were not significantly associated with AKI incidence or severity. Compared with lower Tac IPV, however, very high IPV (>75th percentile) was associated with a significantly increased incidence of grades 2–3 AKI (64.5% vs. 39.2%, *p* = 0.014) and tended to be associated with an increased incidence of severe (grade 3) AKI (22.6% vs. 9.3%, *p* = 0.064). The mCr/bCr ratio was also significantly higher in patients with IPV above than below the 75th percentile (2.54 vs. 2.09, *p* = 0.03). There were no differences in AKI incidence or severity among the three Tac regimens.

Only five patients in this study experienced neurotoxicity and only six experienced thrombotic microangiopathy (TMA) due to Tac. These adverse events were not associated with Tac regimens, Tac IPV, TISS, or C_0_ measurements.

### 2.5. Impact of Tacrolimus IPV, TISS, and C_0_ on Transplant Related Mortality, Relapse Rate, Overall Survival, and Disease-Free Survival

The estimated day +100 and 1-year transplant-related mortality (TRM) rates were 7.0% (95% CI 3.4–12.3%) and 15.4% (95% CI 9.6–22.3%), respectively. The 1-year TRM rate was higher for patients with Tac TISS < 5 ng/mL than TISS ≥ 5 ng/mL (18.2% vs. 7.9%, *p* = 0.18) and for those with IPV above than below the 25th percentile (18.6% vs. 6.1%, *p* = 0.087), although these differences were not statistically significant. The 2-year cumulative relapse rate in the patient’s entire cohort was 20.4% (95% CI 13.7–28.0%). The relapse rate was not affected by Tac IPV, TISS, or C_0_.

Patients were followed up for a median of 18 months (range 0.6–62 months). The 2-year overall survival (OS) rate was 69.7% (95% CI 59.9–77.6%), and the 2-year disease-free survival (DFS) rate was 61.9% (95% CI 52.3–70.0%). Survival outcomes were not affected by Tac IPV, TISS, or C_0_.

## 3. Discussion

This study showed that, in patients receiving PTCy-based GVHD prophylaxis, high Tac IPV during the early period after alloHSCT is an independent risk factor for aGVHD development. To our knowledge, this is the first study showing that Tac IPV could identify patients at greater risk for aGVHD.

The present study, which analyzed the pharmacokinetics of Tac during the first month after allo-HSCT, hypothesized that exposure to Tac and its immunosuppressive effect at that time could affect the recipient’s T cell repertoire and recovery and determine the incidence and severity of aGVHD. Preclinical models have shown that the critical sequence of immunologic events leading to aGVHD occurs within the first few days after transplantation [24]. Fluctuations in Tac C_0_ can be documented during the first month after alloHSCT secondary to concurrent events, such as anemia, mucositis, altered intestinal motility, and use of azoles and other drugs. The present study, therefore, included C_0_ analysis, both as a standard Tac monitoring tool and as a surrogate measurement of Tac IPV.

Tac IPV is an indicator of fluctuations in Tac blood concentrations, identifying episodes of over- and under-immunosuppression that patients can experience over time, thereby reducing the safety and efficacy of Tac. Tac IPV was shown to be clinically relevant in solid organ transplant recipients, but its association with the development of aGVHD had not been assessed in alloHSCT recipients. The present study found that higher Tac IPV was associated with a greater risk of developing aGVHD. Comparisons showed that the risk of aGVHD was 2.5 times higher in patients with IPV above than below the 25th percentile and 3.9 times higher in patients with IPV above than below the 50th percentile. High Tac IPV was also significantly associated with a higher risk of grades III-IV GVHD, despite the low incidence of this complication. Multivariate analysis confirmed that patients with high Tac IPV during the first month after allo-HCT were at increased risk for aGVHD. Moreover, this effect was independent of other known risk factors for this complication.

The incidence of high Tac IPV was significantly lower in patients who received intravenous (31%) than oral Tac, whether BID (47.5%) or QD (51.5%). Reducing Tac IPV should be regarded as a clinical goal, as 82% of patients with Tac IPV reached the C_0_ therapeutic range within 48 h (TISS) of Tac initiation, compared with only 32% of patients with very high Tac IPV. These findings suggest that intravenous Tac should be initiated after alloHSCT, especially in patients with severe mucositis or gastrointestinal events.

Studies have evaluated the optimal range of blood concentrations of Tac early after alloHSCT for preventing GVHD. Although several of these studies reported that the incidence of aGVHD was unaffected by Tac concentrations, more recent studies have shown an association between low Tac levels and a higher risk of aGVHD [15,16,17,25,26]. For example, an analysis of a group of patients who underwent HLA-matched unrelated donor transplantation, most of whom were administered a MAC regimen and all of whom received intravenous Tac and short-term MTX, with a target Tac level set at 10 to 20 ng/mL, found that low mean Tac blood concentrations during the second and third weeks after transplantation were significant risk factors for aGVHD [16]. In patients undergoing RIC alloHSCT using Tac plus MTX, achievements of Tac concentrations >12 ng/mL within the first week after transplant were found to significantly reduce the risk of acute grades II-IV GVHD without impairing the graft-versus-tumor effect [15]. In the latter study, however, lower Tac concentrations at weeks 2, 3, and 4 were not associated with higher rates of GVHD.

Little is known about the relationship between Tac C_0_ and the incidence of aGVHD in patients receiving PTCy. Survival outcomes have been reported similarly in patients with TISS < 10 ng/mL and ≥10 ng/mL early after alloHSCT receiving PTCy-based GVHD prophylaxis [26]. Moreover, TISS < 10 ng/mL was associated with a lower risk of viral infection, with no differences in the bloodstream or fungal infections. Interestingly, TISS < 10 ng/mL was not associated with a higher incidence of GVHD, despite the inclusion of patients with sub-therapeutic TISS (<5 ng/mL). In the present study, low Tac levels during the first week after its initiation (TISS and C_0_ at 7 days <5 ng/mL) were associated with an increased incidence of aGVHD. Moreover, all patients who did not attain the therapeutic range of Tac during that period developed aGVHD.

The main limitations of our study were its retrospective design and that these results may only be applicable to patients undergoing alloHSCT with high-dose PTCy as GVHD prophylaxis. Additional studies are needed to validate these results in alloHSCT using non-PTCy protocols. Nevertheless, to our knowledge, this study is the first to specifically evaluate the impact of Tac IPV on the incidence of aGVHD in patients who underwent alloHSCT.

## 4. Materials and Methods

### 4.1. Patients and Donors

This study, performed at the Hospital Clínic in Barcelona (Barcelona, Spain) between February 2015 and December 2019, retrospectively analyzed data from 128 consecutive patients who underwent their first alloHSCT for malignant hematological diseases using PTCy-based GVHD prophylaxis. Eligibility criteria for transplant included: age 18–69 years, an Eastern Cooperative Oncology Group performance status ≤2, a left ventricular ejection fraction ≥35%, a forced expiratory volume in 1 s and a forced vital capacity ≥40% of predicted, and adequate hepatic function (total bilirubin ≤ 3.0 mg/dL or absence of clinically significant liver disease). Individual data were collected retrospectively by chart review. Only those patients with available information on Tac pharmacokinetic parameters of interest were considered eligible for the study. The protocol was approved by the institutional review board of Hospital Clínic in Barcelona, and all participants provided written informed consent.

### 4.2. Treatment Protocol and Supportive Care

In accordance with institutional protocols, patients were administered specific conditioning regimens based on the type of hematological disease and patient characteristics. Patients aged > 50 years or those who had previously undergone autologous HSCT received a reduced-intensity conditioning (RIC) regimen; otherwise, patients received myeloablative conditioning (MAC) regimens. All patients received fludarabine-based conditioning schemes (detailed in Table 1).

All patients received GVHD prophylaxis, consisting of high-dose PTCy (50 mg/kg IV once daily on days +3 and +4), along with Mesna at 80% of the Cy dose (divided into four doses). Starting on day +5, patients were administered Tac, intravenously (0.03 mg/kg as a 24 h perfusion), orally BID (Tacni^®^, 0.06 mg/kg), or orally QD (Advagraf^®^, 0.12 mg/kg). According to the institutional protocol for GVHD prophylaxis, recipients of haploidentical alloHSCT also received MMF (10 mg/kg every 8 h, maximum 3 g daily) from day +5 to day +35. In addition to these patients, the first nine recipients of HLA mismatched unrelated donor alloHSCT performed in our center with PTCy received Tac plus MMF. After analysis showing that the incidence of GVHD was equal in patients receiving Tac plus MMF and Tac alone, GVHD prophylaxis was modified to include PTCy plus Tac, without MMF [27]. Tac was continued until day +90 and tapered if GVHD grade II-IV was absent. None of the patients in this study received ATG or alemtuzumab for GVHD prevention. Patients undergoing haploHSCT were routinely administered granulocyte-colony stimulating factor (GCSF), starting on day +7 until the absolute neutrophil count reached 1000 cells/mm^3^ for 3 consecutive days.

Antimicrobial prophylaxis was administered according to our institutional practice guidelines. Standard prophylaxis included levofloxacin from day 0 until neutrophil engraftment, fluconazole until day +60, and acyclovir until day +365 (for patients who were seropositive for herpes simplex virus). Standard *Pneumocystis jirovecii* prophylaxis was administered until CD4+ T cell recovery (>200 cells/µL) and/or until immunosuppression was discontinued. The presence of cytomegalovirus (CMV) was assessed weekly by quantitative polymerase chain reaction (PCR) until at least day +60, with pre-emptive therapy initiated if viral reactivation was detected (>1000 IU/mL or two consecutive increases in concentration), according to standard recommendations [28,29].

### 4.3. Laboratory Measurements

Tac C_0_ was assessed by high-performance liquid chromatography-linked tandem mass spectrometry (HPLC-MS) three times per week from the day following the initiation of Tac until the patient was discharged from the hospital discharge and once weekly thereafter. Tac doses were modified to maintain a Tac C_0_ between 5 and 15 ng/mL.

The primary variables of interest for the present study were Tac IPV, therapeutic Tac levels at initial steady state (TISS), and C_0_ at 7, 14, 21, and 28 days after Tac initiation. Tac IPV was estimated using the coefficient of variability (CV). IPV was calculated using the formula: CV-IPV (%) = (standard deviation/mean Tac C_0_) × 100 [22]. TISS was defined as the first serum concentration 48 h after initiation of Tac, a time equivalent to approximately 4–5 times the half-life of Tac (12 h).

### 4.4. Endpoints of the Study

The primary endpoints of this study were (1) to assess the Tac IPV based on C_0_ measurements during the early period after transplant in a cohort of patients who underwent alloHSCT and received PTCy-based GVHD prophylaxis regimens and (2) to analyze the impact of Tac IPV on the incidence of aGVHD. The diagnosis of aGVHD was based on clinical and histopathological findings of affected organs and graded from I to IV according to MAGIC criteria [30].

Secondary endpoints included (1) to analyze the relationship between Tac C_0_ levels as an isolated measurement and the incidence of aGVHD and (2) to analyze Tac-related adverse events, including acute kidney injury (AKI), neurotoxicity and thrombotic microangiopathy (TMA), during the same period of time. AKI was defined according to the 2012 Kidney Disease: Improving Global Outcomes (KDIGO) Clinical Practice Guideline Criteria [30]. Basal (bCr) and maximum (mCr) post-alloHSCT serum creatinine concentrations were measured, with AKI determined according to mCr/bCr ratios determined. The diagnosis and severity of TMA were determined following current recommendations [31,32,33]. The severity of neurotoxicity was evaluated according to the National Cancer Institute Common Terminology Criteria for Adverse Events (AEs) (NCI CTCAE, version 4.0).

### 4.5. Statistical Analysis

The primary endpoint of this study was the cumulative incidence of aGVHD 100 days after alloHSCT. Secondary endpoints were AKI, neurotoxicity, and TMA related to Tac during the first 100 days after alloHSCT.

Categorical variables were reported as numbers and percentages, and continuous variables were reported as medians and ranges. The cumulative incidence of aGVHD was calculated using cumulative incidence methods, with death and relapse included as competing events. Univariate and multivariate regression analyses were performed to assess the effects on aGVHD of the primary variables of interest, including Tac IPV, TISS, type of formulation (intravenous vs. oral), and C_0_ at different time points after transplantation. Other variables that could potentially affect the incidence of aGVHD were included in the statistical analysis, such as age at transplant, sex (male vs. female), disease status at transplant (chemosensitive disease [complete response/partial response] vs. stable disease/progression), CMV donor/recipient status, HLA compatibility (matched related or unrelated donors vs. mismatched unrelated or haploidentical donors), relationship of the sex of the donor to the sex of the recipient (female to male vs. others), and conditioning regimen (MAC vs. RIC). The effect of MMF addition on aGVHD incidence was not analyzed separately from the type of donor, because they were closely related variables. Multivariate regression models included those variables found to be significant in univariate analyses or clinically relevant. All *p*-values were two-sided, with a *p*-value < 0.05 considered statistically significant. All statistical calculations were performed using SPSS Statistics version 25 (SPSS Inc., Chicago, IL, USA) and EZR software.

## 5. Conclusions

For decades, the degree of immunosuppression of alloHSCT recipients has been monitored by measuring Tac concentrations and adjusting doses during the early period after transplant. Despite this, patients continue to present aGVHD, indicating that this method is not entirely effective. The present pilot study suggests that other strategies, such as Tac IPV measurement, in addition to Tac C_0_ determination and dose adjustments, may help identify patients at higher risk for aGVHD. The results of this study showed that, compared with patients with low Tac IPV (<50th percentile), patients with high Tac IPV (>50th percentile) were at significantly higher risks of clinically relevant grades II-IV aGVHD (HR 3.3), including for severe grades III-IV aGVHD (HR 4.99). Patients with high Tac IPV during the first month after transplantation should therefore be closely monitored for aGVHD during the following weeks after discharge from the hospital. Identifying the factors that lead to high Tac IPV is warranted to reduce Tac variability and the risk of aGVHD.

## Figures and Tables

**Figure 1 pharmaceuticals-15-01529-f001:**
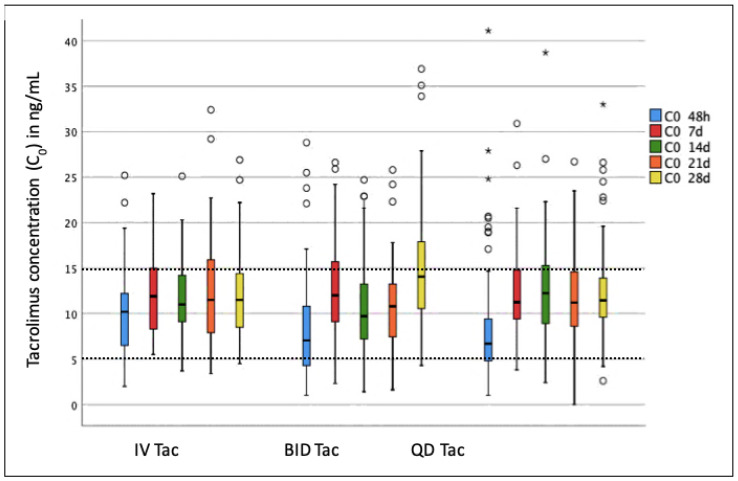
Boxplot graph showing tacrolimus C_0_ measurements at 48 h (TISS) and at 7, 14, 21, and 28 days after Tac initiation according to Tac regimen. Colored boxes correspond to interquartile ranges (IQR), with a dash indicating the median value and the two whiskers corresponding to ±1.5 × IQR. Circles (○) and asterisks (*) show outliers. Dashed lines indicate the therapeutic range.

**Figure 2 pharmaceuticals-15-01529-f002:**
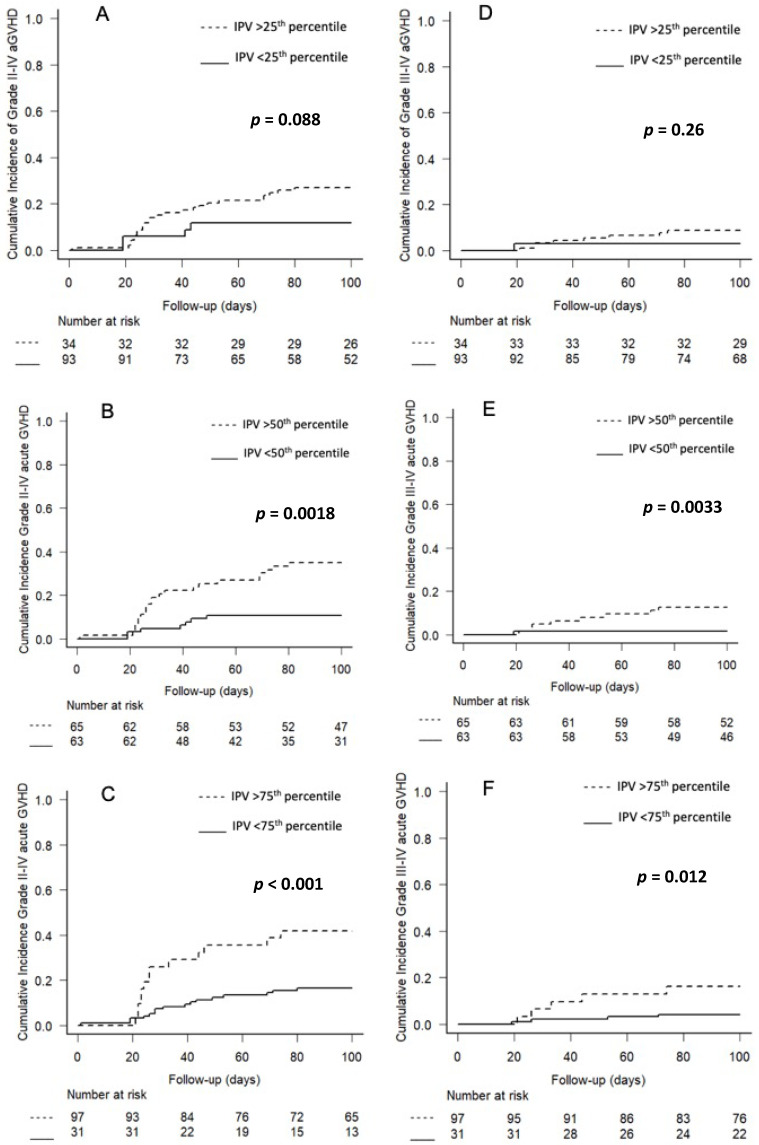
Kaplan–Meier analyses of the effects of Tac IPV on the cumulative rates of grades II-IV and grades III-IV aGVHD. Comparisons of patients with (**A**,**D**) IPV < 25th vs. >25th percentile, (**B**,**E**) IPV < 50th vs. >50th percentile, and (**C**,**F**) IPV < 75th vs. >75th percentile.

**Figure 3 pharmaceuticals-15-01529-f003:**
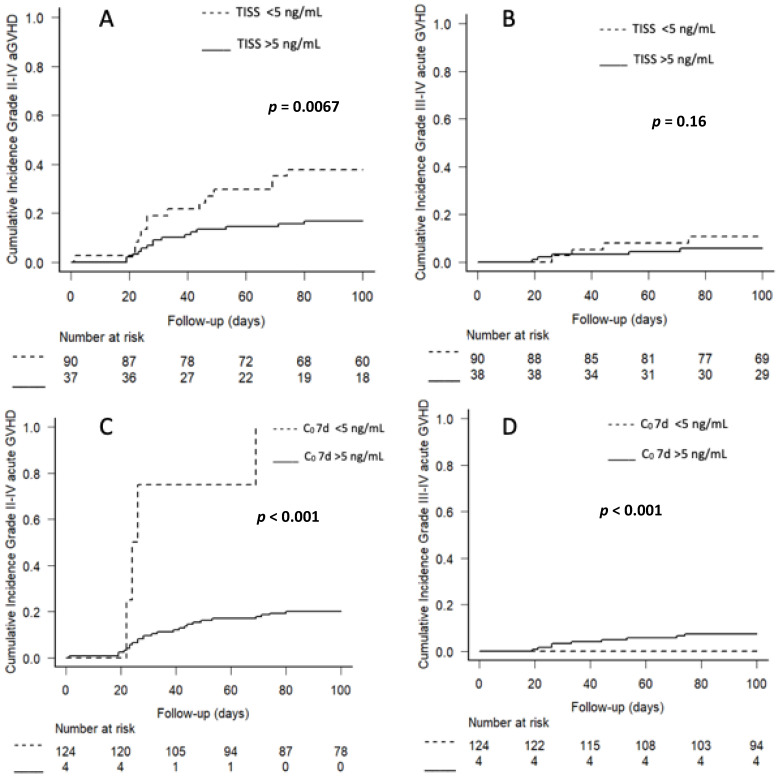
Kaplan–Meier analyses of the effects of (**A**) Tac TISS < 5 ng/mL vs. ≥5 ng/mL and (**B**) Tac C_0_ < 5 ng/mL vs. ≥5 ng/mL at 7 days on the cumulative rates of grade II-IV aGVHD and grade III-IV aGVHD. (**C**,**D**) C_0_ < 5 ng/mL vs. >5 ng/mL.

**Table 1 pharmaceuticals-15-01529-t001:** Demographic and clinical characteristics of alloHSCT recipients and alloHSCT procedures.

Characteristics	All Patients *n* = 128
Age at HSCT, years	
Median (range)	49 (18–69)
Sex (male/female) *n* (%)	72 (56)/56 (44)
HCT Comorbidity Index *n* (%)	
≤2	66 (52)
>2	62 (48)
Primary Diagnosis *n* (%)	
Acute leukemia/Myelodysplastic syndrome	89 (70)
Lymphoma	23 (18)
Multiple myeloma	4 (3)
Chronic mieloproliferative syndromes	11 (8)
Malignant histiocytosis	1 (1)
Disease Status at Transplant *n* (%)	
Complete response	86 (67)
Partial response	25 (20)
Stable disease/Progression of disease	17 (13)
Conditioning Regimen *n* (%)	
Myeloablative	49 (38)
Fludarabine-busulphan	27
Fludarabine-total body irradiation 12 Gy	17
Other	5
Reduced intensity	79 (62)
Fludarabine-busulphan	45
Fludarabine-total body irradiation 8 Gy	27
FLAG-IDA-melphalan	4
Other	3
Donor Type *n* (%)	
HLA 10/10 matched sibling	24 (19)
HLA 5/10 haploidentical sibling	22 (17)
HLA 10/10 matched unrelated	41 (32)
HLA 9/10 mismatched unrelated	41 (32)
Graft Source *n* (%)	
Peripheral blood	125 (98)
Bone marrow	3 (2)
GVHD Prophylaxis Regimen *n* (%)	
PTCy-tacrolimus	94 (73)
PTCy-tacrolimus-MMF	34 (27)
Tacrolimus Formulation at Initiation *n* (%)	
Intravenous	36 (28)
Twice daily oral	32 (25)
Once daily oral modified release	60 (47)
Cytomegalovirus Risk *n* (%)	
Low	18 (14)
Intermediate	70 (55)
High	40 (31)
Pretransplant Renal Function	
Creatinine, mg/dL, median (range)	0.76 (0.39–1.85)

Abbreviations: HSCT, hematopoietic stem cell transplantation; HCT, hematopoietic cell transplantation; G-CSF, granulocyte colony-stimulating factor; GVHD, graft-versus-host disease; HLA, human leukocyte antigen; FLAG-IDA, fludarabine cytarabine idarubicin, and G-CSF; PTCy, high dose post-transplant cyclophosphamide; MMF, mycophenolate mofetil.

**Table 2 pharmaceuticals-15-01529-t002:** Univariate analysis of risk factors for aGVHD.

		Grade II–IV aGVHD HR (95% CI)	*p*	Grade III–IV aGVHD HR (95% CI)	*p*
**Patient Age**	Continuous variable	1.01 (0.98–1.03)	0.56	1.04 (0.99–1.08)	0.07
**Conditioning Regimen**	Myeloablative (vs. RIC)	2.77 (1.36–5.88)	0.005	2.53 (0.72–8.79)	0.15
**Disease Status at Transplant**	Complete remission (vs. other)	0.99 (0.91–1.08)	0.82	0.93 (0.79–1.09)	0.39
**Donor sex**	Female (vs. male)	0.89 (0.43–1.85)	0.76	2.06 (0.43–9.72)	0.36
**Donor Type**	Mismatch (vs. match)	0.67 (0.32–1.39)	0.28	0.80 (0.23–2.83)	0.73
**CMV**	High risk (vs. other)	1.19 (0.56–2.58)	0.65	0.55 (0.11–2.61)	0.45
**Tac formulation**	Oral (vs. intravenous)	1.31 (0.56–3.06)	0.54	1.58 (0.33–7.50)	0.57
**Tac IPV**	>25th percentile (vs. ≤25th) >50th percentile (vs. ≤50th) >75th percentile (vs. ≤75th)	5.53 (0.87–7.35) 3.86 (1.65–9.01) 3.30 (1.62–6.7)	0.088 0.0018 <0.001	3.32 (0.41–26.8) 9.69 (1.20–77.9) 4.99 (1.42–17.49)	0.26 0.033 0.012
**Tac TISS**	<5 ng/mL (vs. ≥5)	2.65 (1.31–5.36)	0.0067	2.39 (0.7–8.16)	0.16
**Tac C_0_ at 7 days**	<5 ng/mL (vs. ≥5)	10.52 (4.88–22.68)	<0.001	*	
**Tac C_0_ at 14 days**	<5 ng/mL (vs. ≥5)	1.01 (0.27–3.79)	0.99	4.30 (0.99–18.72)	0.05
**Tac C_0_ at 21 days**	**	-	-	-	-
**Tac C_0_ at 28 days**	<5 ng/mL (vs. ≥5)	1.08 (0.14–8.28)	0.94	4.96 (0.60–41.01)	0.14

Abbreviations: HR, hazard ratio; RIC, reduced intensity conditioning; CMV, cytomegalovirus; Tac, tacrolimus; IPV, intra-patient variability; TISS, tacrolimus initial steady state; C0, tacrolimus trough concentrations. * All patients (*n* = 4) with Tac C_0_ < 5 ng/mL at 7 days developed grade II aGVHD. ** The number of patients with Tac C_0_ < 5 ng/mL at 21 days was too small for accurate analysis.

**Table 3 pharmaceuticals-15-01529-t003:** Multivariate analysis of risk factors for aGVHD.

		Grade II-IV aGVHD HR (95% CI)	*p*	Grade III-IV aGVHD HR (95% CI)	*p*
**Patient Age**	Continuous variable	1.03 (1.00–1.07)	0.028	1.11 (1.05–1.18)	<0.001
**Conditioning Regimen**	Myeloablative (vs. RIC)	4.46 (1.87–10.65)	<0.001	9.94 (2.84–34.77)	<0.001
**Donor Type**	Mismatch (vs. match)	0.57 (0.27–1.22)	0.15	0.79 (0.20–3.17)	0.75
**Tac IPV**	>50th percentile (vs. ≤50th)	2.99 (1.21–7.39)	0.018	9.12 (1.03–80.56)	0.047
**Tac TISS**	≥5 ng/mL	1.45 (0.65–3.21)	0.36	0.77 (0.22–2.71)	0.69

Abbreviations: HR, hazard ratio; CI, confidence interval; aGVHD, acute graft-versus-host disease; RIC, reduced intensity conditioning; Tac, tacrolimus; IPV, intra-patient variability; TISS, tacrolimus initial steady state.

## Data Availability

All data are included within the article.

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
