# Peer review of "Impact of Early Intrapatient Variability of Tacrolimus Concentrations on the Risk of Graft-Versus-Host Disease after Allogeneic Stem Cell Transplantation Using High-Dose Post-Transplant Cyclophosphamide"

_pharmaceuticals, 2022, doi:10.3390/ph15121529_

Round 1

Reviewer 1 Report

After careful review, I think this manuscript is worth publishing on Pharmaceuticals. The readership can be benefited with information in this manuscript.

Some suggestions are listed below.

1.     As described in the discussion, this is a retrospective study. In section 2.1, the description looks like prospective in nature. The authors have to mention the study design clearly, such as chart review and inclusion and exclusion criteria.

2.     In section 2.2, the authors have to mention why 34 patients additionally received MMF for GVHD prophylaxis. As GVHD is an important endpoint in this study, the authors have to analyze if the use of MMF impacts the outcome.

3.     In section 2.2, the authors have to provide references regarding the indication for CMV preemptive therapy. Besides, the unit of cut-off value for PCR positivity may be mistyped.

4.     In table 1, the authors have better provide more detailed information of some parameters, such as other in primary diagnosis and mismatch unrelated in donor type.

5.     In section 3.2, the authors have better provide a table to summarize the results of Tac IPV, because Tac IPV is an important endpoint of this study.

6.     The figures of Figure 1 and Figure 2 are misplaced, and Figure 2A is not complete. The authors have to indicate the item and the unit of Y axis in Figure 1. In the figure legend, what the circles, boxes, bars, and lines mean should be addressed.

7.     In Figure 2 and Figure 3, the authors have better provide data of grade III-IV aGVHD.

8.     In Table 3, the authors have better analyze data of 25th, 50th, and 75th quartile of Tac IPV separately.

9.     In Line 220, “Table 2” is not correct. It should be “Table 3”.

10.  The authors have better provide a list of abbreviations.

11.  As the instructions for authors, figures and tables should be inserted into the main text close to their first citation. 

Author Response

We would like to thank the editors and reviewers of our manuscript for their comments that have, undoubtedly, helped us to significantly improve this manuscript. We have modified the manuscript according to these comments, with the changes indicated in blue.

    1. As described in the discussion, this is a retrospective study. In section 2.1, the description looks like prospective in nature. The authors have to mention the study design clearly, such as chart review and inclusion and exclusion criteria.

    This study was retrospective study. The inclusion and exclusion criteria are shown in the “Patients and donors” subsection of the Materials and Methods section. This paragraph now reads:

    This study, performed at the Hospital Clínic in Barcelona (Spain) between February 2015 and December 2019, retrospectively analyzed data from 128 consecutive patients who underwent their first alloHSCT for malignant hematological diseases using PTCy-based GVHD prophylaxis. Eligibility criteria for transplant included: age 18–69 years, an Eastern Cooperative Oncology Group performance status ≤2, a left ventricular ejection fraction ≥ 35%, a forced expiratory volume in 1 s and a forced vital capacity ≥40% of predicted, and adequate hepatic function (total bilirubin ≤3.0 mg/dL or absence of clinically significant liver disease). Individual data were collected retrospectively by chart review. Only those patients with available information on Tac pharmacokinetic parameters of interest were considered eligible for the study.

    1. In section 2.2, the authors have to mention why 34 patients additionally received MMF for GVHD prophylaxis. As GVHD is an important endpoint in this study, the authors have to analyze if the use of MMF

    According to previous published data and to our institutional protocol for GVHD prophylaxis, all recipients of haploidentical alloHSCT (n=22) received PTCy + Tac + MMF (n=22). In addition, the first nine patients who underwent HLA mismatched unrelated donor alloHSCT performed in our center with PTCy received Tac plus MMF. GVHD prophylaxis was subsequently modified to include PTCy and Tac after analysis showed that results, including the incidence of GVHD, were identical in patients who received Tac plus MMF and Tac alone (published in Pedraza A. et al, Transplantation and Cell Therapy Journal, 2022).

    Therefore, the effect of MMF addition on aGVHD incidence could not be analyzed separately from the type of donor, because they are closely related variables. As mentioned in “Materials and Methods” section, univariate and multivariate regression analyses explored the impact on aGVHD of the primary variables of interest, such as Tac IPV, TISS, Tac formulation, and C0 at different time points after transplantation, as well as of all other variables that could potentially affect the incidence of aGVHD, such as age at transplant, sex (male vs. female), disease status at transplant (chemosensitive disease [complete response/partial response] vs. stable disease/progression), CMV donor/recipient status, HLA compatibility (match related or unrelated donors vs. mismatched unrelated or haploidentical donor), donor to recipient relation sex (female to male vs. other), and conditioning regimen (MAC vs. RIC). The results of this study showed that, in addition to Tac IPV above the 50th percentile, only age and MAC regimens were associated with a higher risk of aGVHD.

    These issues have been clarified in the “Material and Methods” section.

    1. In section 2.2, the authors have to provide references regarding the indication for CMV preemptive therapy. Besides, the unit of cut-off value for PCR positivity may be mistyped.

    CMV reactivation remains one of the most common and life-threatening infectious complications following alloHSCT. The CMV preemptive treatment strategy is highly effective at managing CMV infection and it is highly recommended by experts in the field. According to published recommendations, CMV preemptive therapy is currently used in our hospital. We have added two of the most relevant references to the manuscript:

    • Ljungman P, de la Cámara R, Robin C, Crocchiolo R, Einsele H, Hill JA, et al. Guidelines for the management of cytomegalovirus infection in patients with haematological malignancies and after stem cell transplantation from the 2017 European Conference on Infections in Leukaemia (ECIL 7). Lancet Infect Dis. 2019; 19(8):e260-e272. doi:1016/S1473-3099(19)30107-0
    • Einsele H, Ljungman P, Boeckh M. How I treat CMV reactivation after allogeneic hematopoietic stem cell transplantation. 2020; 135(19):1619–1629. doi:10.1182/blood.2019000956

    The units of the cut-off value for CMV PCR have been corrected.

    1. In Table 1, the authors have to better provide more detailed information of some parameters, such as “other” in primary diagnosis and “mismatched unrelated” in donor type.

    We have modified Table 1 according to the Reviewer’s suggestions.

    1. In section 3.2., the authors have better provide a table to summarize the results of Tac IPV, because Tac IPV is an important endpoint of this study.

    Thank you for the suggestion. All data concerning Tac IPV are described in the text in Results; so we believe that a table would be redundant.

    1. The figures of Figure 1 and Figure 2 are misplaced, and Figure 2A is not complete. The authors have to indicate the item and the unit of Y axis in Figure A. In the figure legend, what the circles, boxes, bars, and lines means should be addressed.

    These changes have been made, as suggested by the reviewer.

    1. In Figure 2 and Figure 3, the author have better provide data of grade III-IV aGVHD.

    We have added data about grade III-IV aGVHD in Figures 2 and 3.

    1. In Table 3, the authors have better analyzed data of 25th, 50th, and 75th quartile of Tac IPV separately.

    The effects of the 25th, 50th, and 75th percentiles on the incidence of GVHD were analyzed by univariate analysis, with the results shown in Table 2. However, the multivariate analysis cannot include these three variables, as they are mutually exclusive. Multivariate analysis models for each variable (separately) are shown here. The variable that was most significantly associated with both grades II-IV and grades III-IV aGVHD was the 50th percentile.

    Multivariate analysis including IPV 25th percentile.

    Grade II-IV aGVHD

    HR (95% CI)

    P

    Grade III-IV aGVHD

    HR (95% CI)

    P

    Patient Age

    Continuous variable

    1.04 (1.01-1.07)

    0.015

    1.12 (1.05-1.20)

    0.001

    Conditioning Regimen

    Myeloablative (vs. RIC)

    5.37 (2.23-12.95)

    <0.001

    14.55 (4.04-52.39)

    <0.001

    Donor Type

    Mismatch (vs. match)

    0.61 (0.29-1.28)

    0.19

    0.79 (0.21-3.03)

    0.74

    Tac IPV

    >25th quartile (vs. <25th)

    0.58 (0.18-1.82)

    0.35

    0.32 (0.03-3.17)

    0.33

    Tac TISS

    >5 ng/mL

    1.78 (0.79-3.99)

    0.16

    1.11 (0.30-4.00)

    0.88

    Multivariate analysis including IPV 75th percentile.

    Grade II-IV aGVHD

    HR (95% CI)

    P

    Grade III-IV aGVHD

    HR (95% CI)

    P

    Patient Age

    Continuous variable

    1.03 (0.99-1.06)

    0.06

    1.11 (1.03-1.19)

    0.007

    Conditioning Regimen

    Myeloablative (vs. RIC)

    4.63 (1.82-11.78)

    0.003

    11.04 (2.27-53.73)

    0.003

    Donor Type

    Mismatch (vs. match)

    0.64 (0.29-1.36)

    0.25

    0.79 (0.19-3.53)

    0.75

    Tac IPV

    >75th quartile (vs. <75th)

    2.38 (1.08-5.27)

    0.03

    3.70 (0.63-21.72)

    0.15

    Tac TISS

    >5 ng/mL

    1.56 (0.65-3.72)

    0.32

    0.85 (0.18-3.97)

    0.84

    These additional tables can be included as Supplementary Material.

    1. In line 220, Table 2 is not correct. It should be Table 3.

    This line has been corrected.

    1. The authors have better provide a list of abbreviations.

    A list of abbreviations is provided.

    1. As the instruction for authors, figures and tables should be inserted into the main text close to their first citation.

    Figures and tables have been inserted into the main text close to their first citation.

Reviewer 2 Report

This study provided sufficient experimental evidence to support the conclusion.

Overall the results are solid and consistent. Several issues still need to be addressed before publication. 

There are many grammar errors, wrong wording, wrong expression, etc. I suggest a professional expert should edit the Ms before publication.

Author Response

We would like to thank the editors and reviewers of our manuscript for their comments that have, undoubtedly, helped us to significantly improve this manuscript. We have modified the manuscript according to these comments, with the changes indicated in blue.

There are many grammar errors, wrong wording, wrong expression, etc. I suggest a professional expert should edit the Ms before publication.

The manuscript has been thoroughly edited by a native English speaker with scientific expertise.

Reviewer 3 Report

This study by Marco et al analyzes the impact of Tacrolimus intra-patient pharmacokinetic variability and Tacro initial steady levels on the incidence of aGVHD in a cohort of patients who received AlloHSCT followed by PTCy. Overall, this is an interesting study showing that high IPV is associated with higher incidence of aGVHD and higher incidence of AKI without significant associations with NRM. I have the following -in general- minor comments.

1. Despite that the results with the IPV are indeed novel, I am not exactly sure how does the IPV measurement change the current practice as most of the transplant centers treating with tacro measure tacro levels very early and adjust the doses. In other words, how does IPV add something to what we really know re the Tacro levels?

2. It would be interesting to try to identify factors related to high IPV other than the route of administration. This would be an opportunity to identify other modifiable factors to prevent high IPV. The authors can try to run an analysis of the pts' baseline characteristics to see if there is any correlation with IPV or ISS.

3. Was there any correlation between IPV or ISS with incidence of relapse?

Author Response

We would like to thank the editors and reviewers of our manuscript for their comments that have, undoubtedly, helped us to significantly improve this manuscript. We have modified the manuscript according to these comments, with the changes indicated in blue.

  1. Despite that the results with the IPV are indeed novel, I am not exactly sure how does the IPV measurement change the current practice as most of the transplant centers treating with tacro measure tacro levels very early and adjust the doses. In other words, how does IPV add something to what we really know re the Tacro levels?

The Conclusion paragraph has been modified, as suggested by this reviewer:

For decades, the degree of immunosuppression of alloHSCT recipients has been monitored by measuring Tac concentrations and adjusting doses during the early period after transplant. Despite this, patients continue to present aGVHD, indicating that this method is not entirely effective. The present pilot study suggests that other strategies, such as Tac IPV measurement, in addition to Tac C0 determination and dose adjustments, may help identify patients at higher risk for aGVHD. The results of this study showed that, compared with patients with low Tac IPV (<50th percentile), patients with high Tac IPV (>50th percentile) were at significantly higher risks of clinically relevant grades II-IV aGVHD (HR 3.3), including for severe grades III-IV aGVHD (HR 4.99). Patients with high Tac IPV during the first month after transplantation should therefore be closely monitored for aGVHD during the following weeks after discharge from the hospital. Identifying the factors that lead to high Tac IPV is warranted to reduce Tac variability and the risk of aGVHD.

  1. It would be interesting to try to identify factors related to high IPV other than the route of administration. This would be an opportunity to identify other modifiable factors to prevent high IPV. The authors can try to run an analysis of the pts' baseline characteristics to see if there is any correlation with IPV or TISS.

We thank this reviewer for this suggestion. Because Tac IPV based on C0 measurements can be estimated during the early period after stem cell infusion, univariate and multivariate binary regression analyses, rather than a cumulative incidence regression model, can explore potential predictors of high IPV.

The main results of this analysis are summarized in the following tables:

Table S1. Impact of patient’s and transplant’s baseline characteristics on Tac IPV and TISS (univariate analysis).

Univariate analysis

Tac IPV >50th percentile

Odds Ratio (95% CI)

P value

Tac TISS

Odds Ratio (95% CI)

P value

Increasing age

0.98 (0.96-1.01)

0.25

1 (0.98-1.04)

0.49

Sex

  Female (vs. male)

1.55 (0.77-3.12)

0.22

1.94 (0.90-4.18)

0.09

HCT Comorbidity Index

   >2 (vs. <2)

0.87 (0.37-2)

0.74

1.16 (0.48-2.86)

0.75

CMV risk

   High (vs. Low, Intermediate)

0.91 (0.43-1.91)

0.79

0.71 (0.31-1.66)

0.44

Pretransplant Renal Function

   Creatinine, mg/dL

0.21 (0.036-1.24)

0.085

0.54 (0.084-3.53)

0.53

Conditioning Regimen

   RIC (vs. MAC)

1.46 (0.71-3.01)

0.29

1.26 (0.58-2.72)

0.56

Donor Type

   MMUD and haplo (vs. MSD and MUD)

0.64 (0.32-1.29)

0.21

0.71 (0.33-1.54)

0.39

Graft source

   Peripheral blood (vs. BM)

0.51 (0.04-5.74)

0.58

4.94 (0.43-56.24)

0.2

Tacrolimus formulation at initiation

    Oral (vs. IV)

6.44 (2.55-16.26)

<0.001

6.75 (1.93-23.68)

0.003

Table S2. Impact of patient’s and transplant’s baseline characteristics on TISS (multivariate analysis).

Multivariate analysis

TISS

Odds Ratio (95% CI)

P value

Sex

  Female (vs. male)

2.10 (0.94-4.70)

0.072

Tacrolimus formulation at initiation

    Oral (vs. IV)

7.09 (1.99-25.16)

0.002

Because no other baseline characteristics were associated with an increased probability of high Tac IPV (>50th percentile), multivariate analysis was not performed.

The above analysis confirmed the results showing the association between route of Tac administration and the risk of high Tac IPV (>50th percentile) or TISS. A non-significant correlation between female sex and IPSS was documented, with no other association between any of the other baseline characteristics.

These additional data can be included as Supplementary material.

  1. Was there any correlation between IPV or TISS with incidence of relapse?

The 2 year cumulative relapse rate of the entire patient cohort was 20.4% (95% CI 13.7%-28.0%). Relapse rate, however, was not affected by Tac IPV with cutoffs at the 25th (20.3% vs. 21.1%, P=0.85), 50th (20.3% vs. 20.6%, P=0.86) and 75th percentiles (20.4% vs. 20.9%, P=0.91), TISS (<5 ng/mL 20.8% vs. >5 ng/mL 19%, P=0.91); or IPV25th 20.3% vs. 21.1%, P=0.85; IPV50th 20.3% vs. 20.6%, P=0.86; IPV75th 20.4% vs. 20.9%, P=0.91), or C0.

This information has been added to the Results section.

Round 2

Reviewer 2 Report

accept

Reviewer 3 Report

The authors have appropriately addressed my concerns and points.